# Anthropogenic Contamination in the Free Aquifer of the San Luis Potosí Valley

**DOI:** 10.3390/ijerph20126152

**Published:** 2023-06-16

**Authors:** Sonia Torres-Rivera, José Ramón Torres-Hernández, Simón Eduardo Carranco-Lozada, María Elena García-Arreola, Rubén Alfonso López-Doncel, Jesús Anibal Montenegro-Ríos

**Affiliations:** 1Instituto de Geología, Facultad de Ingeniería, Universidad Autónoma de San Luis Potosí (UASLP), Manuel Nava No. 5, Zona Universitaria Poniente, San Luis Potosí 78290, Mexico; sonia.rivera@uaslp.mx (S.T.-R.); jrtorres@uaslp.mx (J.R.T.-H.); maria.garcia@uaslp.mx (M.E.G.-A.); rlopez@uaslp.mx (R.A.L.-D.); anibal.montenegro@uaslp.mx (J.A.M.-R.); 2Instituto Politécnico Nacional, CECyT 15 Diodoro Antunez Echegaray, Dr. Gastón Melo 41, Tenantitla, Milpa Alta, San Antonio Tecómitl 12100, Mexico

**Keywords:** human health, contamination, aquifer, land settlements

## Abstract

The San Luis Potosí valley is an endorheic basin that contains three aquifers: a shallow unconfined aquifer of alluvial material and two deep aquifers, free and confined. The groundwater contamination documented for the shallow aquifer generates contamination of the deep unconfined type aquifer, from which part of the population’s drinking water needs are met. This study records incipient anthropogenic contamination of two types: biogenic and potentially toxic trace elements. The studied contaminants include fecal coliform bacteria, total coliform, nitrate, and potentially toxic elements such as: manganese (Mn), mercury (Hg), arsenic (As), and cadmium (Cd). This contamination in some locations exceeds the permissible limit for human consumption. Some major consequences to health, including severe illness, may be caused by the trace elements. The present results give a first signal about the contamination of the deep unconfined type aquifer due to anthropogenic activity in the valley. This is a priority issue because this aquifer supplies drinking water, and in the short or medium term it will have an effect on public health.

## 1. Introduction

Groundwater is of essential importance to our civilization because it is the largest reservoir of drinking water in the regions inhabited by humans [1]. Although drinking water can be found on the surface in streams or can be extracted from wells, groundwater is preferred because it does not tend to be contaminated with waste or microorganisms. Although groundwater is less polluted than surface water, increasing and diverse human activity due to urban development has gradually accelerated the quantity and diversity of pollutants in wastewater. Contamination sources include urban, mining, and industrial operations. Often, pollutants are beyond the capacity of treatment plants, so untreated water can become contaminated. Contaminated groundwater has thus become a top priority in industrialized countries and must be taken care of [2]. For example, Mushak [3] commented that human beings exposed to chemical and organic pollutants contained in groundwater may be more susceptible to cancer, stomach disease, or malformations in babies. Wang [4] documents that in China, more than 400 cities exploit groundwater—using more than 1/3 of the total water resources—as the only source of water supply, so a series of problems with groundwater utilization has gradually arisen. One of the best-known examples is the contamination with organic solvents and dioxins of the “Love Canal” in New York in 1978, to which high rates of cancer and an alarming number of birth defects have been attributed [5]. In Mexico, agriculture is the human activity that pollutes water the most, but the greatest damage is caused by urban and industrial waste [6]. Some 30 million tons of garbage buried in eastern Mexico City contaminate the region’s groundwater [7]. Diarrhea is the fifth-leading cause of infant death in Mexico and is associated with the water consumed [7]. In the future, it is expected that groundwater sources will increasingly become the only source of additional water for community development, particularly in more remote areas. However, the quality and quantity of these resources are constantly threatened by the activities inherent to the growth and development of urban centers. In the San Luis Potosí Valley (VSLP), recent water quality studies documented that the water contains high concentrations of fluoride (attributed to the substrate draining surface water and water–rock interaction) and As (assigned to mining activity [8]) in addition to other elements derived from anthropogenic activity. Currently, anthropogenic activities have a high impact on the quality of groundwater in the VSLP (housed in three aquifers—one unconfined and two deep aquifers) with the increase in population, change in land use, and expansion of industrial and mining activities. These factors were responsible for the contamination of the unconfined aquifer with heavy metals, major ions, nitrates, fats, oils, grease, and bacteriological contamination. The low average annual precipitation does not contribute substantially to surface water recharge, but it is considered that in times of rain, the contribution is by infiltration in the margins of the valley that mainly feed the unconfined aquifer and the deep unconfined type [9] (Figure 1b). Aguirre and Martinez [10] determined the areas of the valley where pronounced depletion cones are manifested by the pumping of deep wells, and they point out that this highlights the over-exploitation of this aquifer. Another important factor to examine is the subsurface geology of the valley. Aguirre [11], based on the information of lithological sections from deep wells, documents the structural and geological situation of the subsoil of the VSLP and Villa de Reyes (Figure 1a). With this, the author verified the existence of basement blocks delimited by staggered faults along with their course and range of settlement, thicknesses, and continuity of the lithological units of the subsoil, which are fundamental data to define the dimensions and real geometry of the deep aquifer of that valley. The “Atlas de riesgo” for the municipalities of San Luis Potosí (capital) and Soledad de Graciano Sanchez documents land settlements (Figure 1a) that coincide with the limits of some of the basement blocks in the subsoil reported by [11], suggesting that the settlements are controlled by these structures. The water-quality records where contamination was detected in the shallow aquifer (unconfined type) coincide with the surface areas where ground settlements and faulting (with rupture of the hardened soil substrate or tepetate that supports it) have been reported in the Valley of San Luis Potosí [12]. This paper argues that the widely documented anthropogenic contamination of the shallow aquifer [8,9,13] is manifesting, incipiently, in one of the two deep aquifers (unconfined type) and is linked to the possible deterioration and rupture of drainage pipes (especially old drainage pipes) because the settlement and faulting of the valley terrain are creating conduits for percolation of contaminated water into the deep unconfined aquifer. The importance of these studies is that a large part of the population of the municipalities of San Luis Potosí capital; Soledad of Graciano Sanchez, S.L.P.; and Mezquitic de Carmona, S.L.P.; receive water for human consumption from the aquifer examined here, so it is a priority to evaluate the risks that this implies for the health of the population and to take the necessary measures to prevent them. As Espinoza [14] suggests, the infiltration of contaminated water can significantly affect changes in the chemical composition and can cause extreme cases of remediation costs in the drainage network, treatment plants, etc.

This work documents that contamination by trace elements—with potentially harmful effects on health—have percolated from the contaminated shallow aquifer to the deep unconfined type. This process is assumed to be of a temporary and recent nature due to ground settlement processes and fracturing of the tepetate layer that supports the shallow aquifer.

## 2. Materials and Methods

According to the topography of the study area, the recharge and discharge zones were identified and selection of the points of wells and waterwheels was carried out. The samples of wells that correspond to the deep unconfined type have an average depth of 260 m. In the shallow aquifer, wells have a depth of approximately 40 m [9]. Sample bottles were washed with 10% hydrochloric acid (HCl) and phosphate-free detergent, rinsed more than three times with distilled water, and allowed to dry. In the hydrogeochemical sampling campaign, 29 groundwater samples were taken from 22 wells and 7 waterwheels. Each sample was taken using a peristaltic pump with a hose with two ends; the sample was passed through one end, and the other end was passed through a 0.45 μm filter and deposited into the duly labeled polyethylene container. In situ physical parameters such as pH, T (temperature), EC (electrical conductivity), DO (dissolved oxygen), ORP (oxide reduction potential), and total dissolved solids (TDS) were measured with a Hanna HI 9829 multiparameter, and sample alkalinity was tested with an HI3811 test kit with a 10 mL phenolphthalein indicator, 10 mL bromophenol blue indicator, and 120 mL alkalinity titrator. Samples taken for cation and trace element testing were preserved with ultra-pure concentrated nitric acid and kept in a cooler for transfer to the laboratory [15,16].

### 2.1. Location

The state of San Luis Potosí is located north of the Central Plateau in the central–eastern part of Mexico between the parallels 21°09${}^{\prime }$35${}^{\prime \prime }$ and 24°33${}^{\prime }$09${}^{\prime \prime }$ north latitude and the meridians 98°19${}^{\prime }$52${}^{\prime \prime \prime }$ and 102°17${}^{\prime }$51${}^{\prime \prime }$ west of Greenwich (Figure 2). It is divided into four natural regions: Altiplano, Centro, Zona Media, and Huasteca [17]. It is divided into two contrasting hydrogeological regions: The Salado region, located in the north–central portion of the state and with an area of 35,164.19 km${}^{2}$, presents a dendritic endorheic hydrological pattern where surface water currents are intermittent, scarce, and with little flow—flowing only during the rainy season in summer and occasionally in winter. The second is the Pánuco region, located in the south–southeast portion of the state and has an area of 27,140.55 km${}^{2}$ and a dense fluvial network of perennial streams that form some rivers that flow into the Gulf of Mexico [18]. The San Luis Potosí aquifer is defined with the code 2411 in the Geographic Information System for Groundwater Management (SIGMAS) of the CONAGUA [19] (Figure 2).

There are three aquifers in the San Luis Potosí Valley (VSLP). One is of the unconfined type and has a shallow depth of only approximately 40 m [19] and a stratum of low hydraulic conductivity; it is recharged from precipitation in the valley, and its behavior is very dynamic. It has diffuse induced recharge (occasionally by drinking water and/or drainage leaks) and by irrigation water returns [9]. Irrigation return flows are responsible for the deterioration of groundwater quality in a large number of countries, particularly in semiarid and arid regions [20]. Several studies have detected the presence of organic and bacteriological contaminants in the water captured by this aquifer, which has been attributed to the shallow depth at which it is located and the fact that most of the crops are irrigated with untreated municipal sewage; in addition, contamination by some metals can be linked to anthropogenic activities [13,21,22,23]. The aquifer material has textural variations that influence its characteristics as an aquifer. Towards the Sierra de San Miguelito (SSM), conglomerates immersed in a sandy-clay matrix predominate, and towards the northeast of the east valley, silts and sands predominate. The direction of flow of the unconfined aquifer is from southwest to northeast [24], and its recharge depends on conditions such as precipitation, evapotranspiration, runoff, vegetation, presence of soil layer, the slope of the terrain, and permeability of the material. The main recharge in the aquifer takes place in the southwestern and western flank of the VSLP from the streams that flow down from the SSM towards the valley [25]. In this region, the boundary of the unconfined aquifer is located towards the La Palma locality, where the granular material is wedged and its contact with the fractured volcanic material is closer. The aquifer of the granular medium feeds the wells and rests on the hardened soil or “tepetate”. The other unconfined type (and deeper) aquifer, which is hosted in the granular alluvial material that fills the tectonic trenches that are now buried by the valley fill, has been the most exploited. Geological structures such as faults control the distribution and thickness of this aquifer unit, which is composed mainly of Paleogene and Quaternary clastic materials. The aquifer is currently exploited by wells that reach depths of up to 350 m of sedimentary material. Its upper limit is approximately 100 to 150 m deep. It is confined in the center of the valley by a sedimentary layer that is not very permeable. Its thickness ranges from 100 to 200 m [23]. Another aquifer of the confined and deeper type is hosted in alluvial material that was covered by a pyroclastic unit (an ignimbrite) that constitutes its seal and separates it from the unconfined type. Most of the wells drilled in this aquifer have depths of 350–450 m, although there are some that are 800–1000 m [25]. The depth of the potentiometric surface is greater than 150 m depending on the location within the valley.

### 2.2. Geology

The aquifer studied is located in an endorheic tectonic valley whose lowest topographic parts are in its northeastern part. The various studies carried out to date [11,25,26] have fully established the geology around the valley, which is outlined below. The basement is a sequence of calcareous and calcareous–marine clay–calcareous rocks (basin and platform) of Mesozoic age, outcropping in the Sierras of Alvarez and El Coro, which limit the valley in the eastern and northeastern parts. The north, west, and southwest limits are: La Melada and San Miguelito mountains (respectively), which are formed of Paleogene (Oligocene–Miocene) volcanic rocks of rhyolitic and dacitic composition. The valley was formed by extensional faults in two main systems, those of NW-SE orientation (north and southwest limits of the valley) and those of N-S orientation in the other limits. Geological events that shaped the valley generated a series of tectonic blocks that remained under the alluvial fill and have been determined by the coincidence of lithological columns cut by numerous wells at different depths and evidenced by magnetometry geophysical studies [27]. Thus, the aquifers have been documented: the deep confined in clastic deposits between the Mesozoic basement and pyroclastic deposits (ignimbrites) from the middle Oligocene that outcrop regionally; intermediate in the fill of alluvial material deposited on the ignimbrites; and upper “pendant” (5–30 m higher) that has its base on the hardened layer called “tepetate”, which is typical of the arid and semi-arid regions of Mexico. Due to its characteristics, this shallow aquifer is the most susceptible to being impacted by anthropic activity and is the object of this research study, especially with respect to its influence on the incipient contamination of the deep unconfined type.

### 2.3. Sample Analysis

The analysis of the samples was carried out at the geochemistry laboratory of the Geology Institute, Universidad Autonoma of San Luis Potosí. The analysis consisted of determining the concentrations of the major ions sodium (Na${}^{+}$), potassium ($K^{+}$), calcium (Ca${}^{+}$${}^{2}$), and magnesium (Mg${}^{+}$${}^{2}$)] using an optical emission spectrophotometer with plasma coupled to atomic absorption induction (ICO-OES). Induction-coupled plasma–inductively coupled mass spectrometer (ICP-MS) was used to determine trace elements such as mercury (Hg), barium (Ba), strontium (Sr), cadmium (Cd), lead (Pb), silver (Ag), rubidium (Rb), cobalt (Co), copper (Cu), iron (Fe), arsenic (As), lithium (Li), nickel (Ni), manganese (Mn), chromium (Cr), zinc (Zn), and aluminum (Al). The UV-Vis method with HACH DR/2000 equipment was used for the analysis of nitrates $\left(N-NO_{3}\right)$, sulfates ($SO_{4}$${}^{-2}$), and fluorine ($F^{-}$); the volumetric method tested for bicarbonates ($HCO_{3}$${}^{-}$) and carbonates (CO${}_{3}$${}^{-2}$); the argentometric method tested for chlorides ($Cl^{-}$); the analysis of pH, electrical conductivity, alkalinity, total dissolved solids, fats and oils (GA), total coliforms (CT), and fecal coliforms (CF) was performed in the groundwater laboratory of the Faculty of Engineering of the UASLP.

## 3. Results

In February 2020, the sampling campaign of the VSLP was carried out in order to obtain its physicochemical parameters in situ (Table 1), and, in addition, the characterization of the chemical composition of the groundwater of the valley was carried out in Table 2. These parameters are expressed for a 10.57 km physicochemical section (Figure 3 of the VSLP. It begins in the southwestern part of the valley with sample 3P located along Hernán Cortés avenue; crosses the valley towards the center of the city with samples 1P, 10P, and 9P; heads north of the valley for 4N, 6P, 8N, and 7P; and at the end of the stretch is sample 2P—this development is the furthest from the VSLP, with the volcanic rocks at the base of this aquifer approximately 360 m deep. The pH at the beginning in the 3P sample was 7.7; it increases and reaches values of 8 in 5P and maintains little variation until Tangamanga park; however, in the end, values of 8.54 are reached in 2P (Figure 3 and Table 1). The temperature starts at 38 °C, and throughout the section, the temperature is discontinuous and decreases gradually until reaching 26 °C at the end of the section in the El Saucito 2P sample (Figure 3 and Table 1). The average electrical conductivity in the water is 488 μS/cm in the entire section. It starts low at 3P (769 μS/cm); the highest points are located almost in the center of the section in the valley at 4N (808 μS/cm) and 8N (930 μS/cm) due to the chemical components dissolved in the water; the value decreases at the end of the section in the 2P samples, which suggests that the waters interact with volcanic rocks (Figure 3 and Table 1). Alkalinity starts with values of 156 mg/L at 3P, increases slightly towards the center of the valley to 210 mg/L at 9P and 10P, and increases to 348 mg/L at 4N and 8N; this increase is associated with the interaction of water with carbonates in the sands and silts of the fill material (Figure 3 and Table 1). Regarding the total dissolved solids (TDS), this value begins with a concentration in 3P of 206 mg/L, increases towards the center of the valley in 1P (391 mg/L), 4N (403 mg/L), and 8N (465 mg/L), and decreases at the end. The concentration changes in 4N and 8N that are in the shallow aquifer reflect the contamination of the shallow aquifer; added to the above, this is consistent with the CE ion network and the salts, minerals, metals and any other organic or inorganic compounds (Figure 3 and Table 1).

The oxidation/reduction potential in units of milliVolts (mV) starts low at 3P (209 mV), the high values are represented at 5P (284 mV), 4N (210 mV), and 8N (312 mV), and the last point concludes with a low concentration in 2P (132 mV) (Figure 3 and Table 1). The samples analyzed in the laboratory must comply with the principle of electroneutrality: the sum of the charges of all the cations must be equal to the sum of the charges of all the anions and they must be electrically neutral [28]. To check the accuracy of the analysis of the majority ions, ionic balance is performed (Equation (Equation 1)).
(1)$$\%Electroneutralidad=\frac{\sum Cations-\sum Anions}{\sum Cations+\sum Anions}\times 100$$

Cations and anions are expressed in meq/L [29]. For the analysis of a sample to be valid, the amplitude of variation of the electroneutrality percentage must be ±5%, although an amplitude of up to ±10% can be accepted [30]. In this case, the interval ±10% was considered (Table 2). The predominant water family of the analyzed samples presents 55.2% of bicarbonate-calcium composition ($Ca-HCO_{3}^{-}$), bicarbonate-sodium ($Na-HCO_{3}^{-}$) is second place and represents 37.9% of the total samples, and 6.89% were of chloride-calcium composition ($Ca-Cl^{-}$) (Table 2) in coherence with the nature of the fractured and carbonate volcanic rock of the main permeable materials existing in the center of the VSLP.

The aforementioned water families reflect that the main supply corresponds to the Santiago River, and the salt content corresponds to the interaction of bicarbonate-sodium water where the water movement is in contact with the fractured volcanic material, sand, and silt; so the flow of this may have hydraulic communication with the aquifer of the granular medium that feeds the wells and rests on the hardened soil or “tepetate”. The bicarbonate-calcium family may represent the interaction of the water with the granular material that filled the tectonic pits, and the chloride-calcium family may be due to the agricultural activity developed in that area. The chemical composition of groundwater is the result of continuous processes of interaction between the meteoric water that infiltrates the ground and circulates through the different subsoil materials. Chemical analyses of water samples taken directly from the wells—plotted on a Piper diagram [31]—define different (types of) water families (Figure 4). The bicarbonate-calcium family is represented by the pink diamond, is located in the east and center of the VSLP, and indicates samples that passed through carbonate rocks (1P, 2P, 3P, 7P, 11P, 12P, 13P, 14P, 16P, 19P, 20P, 21P, and 22P); 4N, 23N, 26N and 27N are the result of processes of water mixtures found in the center of the diamond, and samples with a direction towards the south of the gray diamond are also observed: this represents an ion exchange corresponding to samples 3P, 5P, 6P, 10P,15P, 17P, 28P, and 29P; 8N, 24N and 25N are from deep wells, and only two chloride-calcium samples were found (9P and 18P), which correspond, respectively, to 280 m and 350 m depths that cut through granular material that filled the valley; well 9P is located in the center of the city of San Luis Potosí, and well 18P is to the NE of the valley in the Agronomy School (Figure 4).

Ionic ratios are used to support the identification of possible groundwater origins or mixing processes occurring in the subsurface; the $Na^{+2}K^{+}$ vs. $Cl^{-}+SO_{4}^{-2}$ ratio is proposed by [32] to identify the presence of regional flows as well as in the characterization of some evolutionary processes of geogenic or anthropogenic origin. Thus, it was used in the identification of these for the study area (Figure 5). In this study, most of the samples were found in the local flow area, and five were found in the intermediate flow, which means that the water is of recent infiltration and that it has had little chemical evolution and water–rock interaction.

Table 3 shows the results of nitrogen and phosphorus contents of the sampled works (wells and waterwheels) and localities referred to in the maps. Table 4 shows the results of bacteriological contents, and Table 5 and Table 6 show the contents of trace elements and metals, respectively.

### 3.1. Nitrogen and Phosphorus Content

Sample 8N (Table 3) is the only one with anomalous nitrogen content and belongs to a 3 m deep private well located in settlement 3a Chica in the El Saucito zone north of the VSLP. It presents a concentration of 15 mg/L of nitrate nitrogen ($N-NO_{3}^{-}$), which exceeds the maximum allowable limit of 10 mg/L set by NOM-127-SSA1-1994 [33].

### 3.2. Bacteriological Results

Regarding the biogenic content in the sampled waters related to anthropogenic activity, the results are shown in Table 4. Of the seven wells sampled, only three show high biogenic contents related to anthropogenic activity. These are noria 4N, with a concentration of 2 NMP/100, and norias 24N and 27N, with concentrations of 11 NMP/100 and 5NMP/100, respectively. Concerning the wells, the presence of fecal coliforms is documented in wells of the deep unconfined type of the VSLP. Of the 29 wells studied (100%), 12 samples (41%) were found to be out of the norm. These are: well 2P with a concentration of 3 NMP/100 (most probable number per 100 mL); well 3P with a concentration of 8 NMP/100 (from Morales Park); well 12P with a concentration of 11 NMP/100; wells 15P and 16P with concentrations of 11 NMP/100; well 17P with a concentration of 14 NMP/100; wells 18P and 19P with concentrations of 5 NMP/100 and 17 NMP/100, respectively; well 20P with a concentration of 11 NMP/100; well 22P with a concentration of 5 NMP/100; and well 28P with a concentration of 11 NMP/100.

### 3.3. Fats and Oils

The measurement of fats and oils does not measure a specific substance but a group of substances with the same physicochemical characteristics (solubility). Therefore, the measurement of fats and oils includes fatty acids, soaps, fats, waxes, hydrocarbons, oils, and any other substance that can be extracted with hexane, and its permissible limit is 26 mg/L according to NOM-002- SEMARNAT-1996 [34]. Therefore, almost all the samples show low concentrations of fats and oils, although they present small amounts.

### 3.4. Metals and Trace Elements

Table 5 and Table 6, respectively, document the contents of metals and trace elements measured in the samples studied. In these tables, we highlight that the out-of-range samples correspond to Mg, As, Cd, and Hg. For Mg, only one sample exceeds this permissible limit: well 18P with a concentration of 0.84 mg/L. For As, wells 24N (in Soledad) with a concentration of 0.05 mg/L, 12P with a concentration of 0.126 mg/L, and 18P with a concentration of 0.09 mg/L are the ones with contents that exceed the permitted levels (Figure 6a).

For Cd, the samples that exceed the limit are: wells 24N and 26N with concentrations of 0.0067 mg/L and 0.0119 mg/L, respectively; 11P with a concentration of 0.0169 mg/L; 12P with a concentration of 0.0206 mg/L; P14 with a concentration of 0.005 mg/L; 17P with a concentration of 0.0085 mg/L; 20P with a concentration of 0.01068 mg/L; and 29P with a concentration of 0.00524 mg/L (Figure 6b). For Hg, the samples that fall outside the norm are: wells 23N with a concentration of 0.004 mg/L; 25N with a concentration of 0.002 mg/L; 11P with a contraction of 0.002 mg/L; 18P with a concentration of 0.0015 mg/L; 20P with a concentration of 0.002 mg/L; and 22P with a concentration of 0.0012 mg/L (Figure 6c).

## 4. Discussion

Moran [13] and Lopez [9] studied water quality in this shallow (unconfined) aquifer and detected, in general, significant levels of nitrates, sulfates, chlorides, and, in the urban zone, punctual anomalies of heavy metals (mercury, barium, strontium, cadmium, lead, phosphorus, and silver). These authors report the highest anomalies in heavy metals for the industrial zone and in places where irrigation is done with recycled water from a water storage tank from the industrial zone. The distribution of the wells sampled in these studies coincides, roughly, with the distribution of the ground settlement faults that have been documented in the valley [12], which suggests that if the lower boundary of this aquifer has been fractured, water contaminated with these heavy metals is possibly infiltrating into the deep unconfined type housed in the granular medium that fills the valley. The results of the sampling and analysis of water from 22 wells and 7 waterwheels in this study show anthropogenic contamination. The anthropogenic contaminants (fecal coliforms, total coliforms, and trace elements) have a direct relationship to the drainage systems, and the sites where contents exceeding the permissible limits for human consumption set by NOM-127-SSA1-1994 [33] and NOM-002-ECOL [34] were detected in groundwater may indicate areas where there is advanced deterioration of the drainage pipes and which also have conduits for percolation to depth. In the El Saucito area, one of the settlement structures that has been tracked for nearly 30 years affects a large number of settlements north of the city, some of which have been urbanized by the use of very sensitive drainage materials. Faults in the event of subsidence are a serious source of contamination of shallow suspended aquifers, and, according to these results, cause infiltration of deep aquifers without limits. On the other hand, the Santiago River, with a distance of about 12.7 km [35], is the natural bed of the city and historically represents the natural edge that limited the growth of the urban sprawl; since flooding causes road safety and environmental health problems, the fracturing of the land within this river plays an important role during flooding. During the rainy season, part of the water infiltrates into the San Miguelito mountain range, and other surface water flows down and feeds the Santiago River as it follows its course from east to northeast, connecting with the rest of the territory and the wastewater treatment plant of the Tenorio Tank located in the industrial zone, then joining in a single flow to the northeast, passing by the side of the Agronomy School where well 19P and well 18P are located. These wells are contaminated with fecal coliforms and total coliforms, and the wells located in Soledad also have anthropogenic contamination. It is safe to assume that both factors—land settlement and the rupture of old drainage systems—may be contributing to the contamination of two aquifers in the San Luis Potosí Valley. In this case, the ground settlement structures that fracture the cemented and hardened material that supports the shallow unconfined aquifer are more towards the central–eastern part of the valley, which could facilitate the infiltration of contaminated water towards depth.

### Nitrogen and Phosphorus

Nitrate and nitrite ions occur in soils and water as part of the nitrogen cycle in the earth. Nitrate constitutes the major total amount of nitrogen available in surface waters. Nitrogen occurs naturally in soils and is typically bound to organic and mineral matter in the soil. Life depends, among other things, on the proportion of nitrogen (*N*) and phosphorus (*P*) that is available in the medium. Normally, there is much more nitrogen than phosphorus [36]. However, at concentrations that exceed the norm of nitrates, they can cause methemoglobin, that is, oxygen deficiency in the blood, causing death. In this study area, only one sample (8N) presented a concentration of 15 mg/L, as shown in Table 3. Biogenic contaminants have a clear relationship with human wastes that are normally channeled through the drainage system and are driven to their dispersion towards the NE part of the VSLP in the cultivated areas in that sector of the valley. In this sense, the dispersion of organic pollutants should be preferentially manifested towards that part; nevertheless, the locations and the values found in the wells of the central part—where most of the population and commercial activity is concentrated—allow us to suppose that there is also percolation of drainage wastes towards the unconfined aquifer, and that this occurs due to rupture of the drainage pipes. A major problem with this situation lies in the possibility that this biogenic contamination penetrates the deep unconfined type because part of the distribution of drinking water comes from that aquifer. This possibility has been confirmed in this study in at least one of the wells (2P), which contains a high concentration of total coliforms (up to 150 NMP/100), and, according to NOM-127-SSA1-1994 [33], the permissible limit for total coliform organisms in water for human consumption is 2 NMP/100 (most probable number per 100 mL); however, most of the wells sampled have contents of <3 NMP/100. This contamination is already reflected as increasing in 13 more wells (values between 5 and 16 NMP/100; see Table 4) covering a very wide area. The contrast of the area where this type of contamination with a clear anthropomorphic relationship is detected with the system of ground settlement faults in the VSLP suggests that both phenomena have a connection, given that this differential subsidence is linked to rupture of the “tepetate” layer that supports the shallow unconfined type aquifer that is known to be contaminated [9,13]. The percolation conduits to the deep aquifer are constituted by these faults. The well with a high value of these contaminants is located in an area where the greatest development of ground settlement has been detected: the most representative evolution of which is the Aeropuerto fault, which has been monitored for more than 20 years. Table 4 presents the results of total coliforms in the wells and waterwheels studied in the VSLP. However, 13 more samples (44%) have concentrations between 5 and 16; 8 of these (wells 15P, 16P, 17P, 18P, 19P, 20P, and 22P) have concentrations between 11 and 17 NMP/100. The other 15 samples have <3 NMP/100. The bacteriological results of the studied samples reflect the low quality of water for supplying the population in the valley. As for the wells, well 24N has a concentration of 11 NMP/100, and well 27N has a concentration of 5NMP/100; both are located in Soledad. PExceeding the allowable limit of fecal coliforms with no detectable NMP/100 mL, for example, in well 3P with a concentration of 8 NMP/100,are exceeded, for example, in well 3P with a concentration of 8 NMP/100, well 12P with a concentration of 13 NMP/100, wells 15P and 16P with concentrations of 11 NMP/100 and 16 NMP/100, respectively, well 17P with a concentration of 14 NMP/100, well 18P with a concentration 10 NMP/100, well 19P with a concentration of 17 NMP/100, well 20P with a concentration 16 NMP/100, well 22P with a concentration of 7 NMP/100, and well 28P with a concentration of 11 NMP/100. (Table 4). The other 15 samples have <3 NMP/100. The bacteriological results reflect the low quality of water for supplying the population in the valley. The noria 24N has a concentration of 11 NMP/100, and the noria 27N has a concentration of 5NMP/100. Concerning fats and oils, their detected content varies in value from 2.65 to 7.7, and the concentration increases in the wells towards the center of the valley following the direction of the subway flow. Of the eight wells where fats and oils were detected in the water, all have low concentrations (between 2.65 and 7.70 mg/L) with regard to the NOM-002-SEMARNAT-1996 [34] standard, which establishes 26 mg/L as the permissible limit. Only four of the water samples have concentrations above 6.2 mg/L, but all of them show contamination in the process; because fats and oils are a product of anthropogenic activity, the fact that they are present in the wells reveals that they have percolated into the deep aquifer. This is supported by two important factors: (1) The central zone of the VSLP has the highest concentration of ground settlement faults. This is logical since in the center of the city of San Luis Potosí there are restaurants, hotels, public bathrooms, the train station, etc. (2) The concentration increases towards the wells in the center following the direction of the subway flow. Another factor to consider is the presence of trace elements (with some heavy metals) in the wells of the deep unconfined type. The values detected are presented in Table 5 and Table 6. Manganese (Mn) is considered a mineral associated with igneous and metamorphic rocks containing divalent Mn as a minor constituent; in particular, it is significant in basalt due to its dominant mineralogy of olivines, pyroxene, and amphibole. Small amounts are present in dolomites and limestones in place of calcium [37]. The previous source is mainly responsible for the contribution of *Mn* in the study area. This element is normally found in organisms as an activator of certain enzymes. When ingested in large doses, it is a poison that mainly affects the central nervous system; in appreciable quantities, it produces an unpleasant taste in the water, which makes its presence noticeable when drinking and its toxic action more easily avoided [37]. The permissible limit for human consumption according to NOM-127 is 0.15 mg/L; one sample exceeds this permissible limit: well 18P with a concentration of 0.84 mg/L (Figure 6a). Arsenic is important in water chemistry, especially since the modern use of pesticides has become widespread and these products contain this element. As is found free in nature as a steel-gray, brittle solid [38]. High concentrations of arsenic are commonly associated with sediments that are partially derived from volcanic rocks of acidic or intermediate composition [29]. The presence of this element is most likely due to the occasional input of arsenic-containing fertilizers; however, it may also be due to the dissolution of arsenic-bearing volcanic rocks within the study area. Long-term exposure to As via drinking water at concentrations of 0.05 mg/L and even lower causes skin, lung, bladder, and kidney cancer and skin alterations such as pigmentation changes and thinning of the skin. Immediate symptoms of acute poisoning include vomiting, abdominal pain, and hemorrhagic diarrhea [39]. The permissible limit established for arsenic (As) by NOM-127-SSA1-1994 [33] is 0.05 mg/L. The out-of-range samples are: the 24N Soledad waterwheel with a concentration of 0.05 mg/L and wells 12P with a concentration of 0.126 mg/L and 18P with a concentration of 0.09 mg/L (Figure 6a). Cadmium is considered, biologically, neither beneficial nor essential for man, but it is a toxicant that acts on the kidneys and liver and produces nausea and vomiting. Cadmium poisoning produces arterial hypertension, and it is a proven carcinogen [38]. As for its presence in groundwater, its probable source is of external origin, and it is not so much associated with the dissolution of minerals containing cadmium in their composition. Water contaminated by cadmium generates corrosion of the pipes used because cadmium is a contaminant of galvanized iron and zinc [38]. The permissible limit for cadmium (Cd) in human consumption is 0.005 mg/L according to NOM-127-SSA1-1994 [33]; the samples that exceed the limit are the waterwheels 24N with a concentration of 0.0067 mg/L and 26N with a concentration of 0.0119 mg/L and wells 11P with a concentration of 0.0169 mg/L, 12P with a concentration of 0.0206 mg/L, 14P with a concentration of 0.005 mg/L, 17P El Ranchito with a concentration of 0.0085 mg/L, 20P with a concentration of 0.01068 mg/L, and 29P with a concentration of 0.00524 mg/L (Figure 6b). These values indicate that for the shallow aquifer—the zone where the 24N well are located—contamination by this metal is only slightly higher than the standard (0.0067 mg/L), while for 26N it is double this value (0.0119 mg/L). Wells 14P and 29P show incipient values, or rather, values at the limit of the standard (0.005 and 0.00524 mg/L, respectively). Wells 11P, 12P, and 20P show values two to four times the norm (0.01068, 0.01690, and 0.02060 mg/L, respectively), which is highly worrisome. Mercury is one of the most widely distributed metals in the environment and is known for its high toxicity (mainly methylmercury); it is harmful to the environment and can bioaccumulate. It can come from both natural and anthropogenic sources. It is released naturally by mobilization generated in the earth’s crust, by volcanic activity, and by rock erosion. Anthropogenic sources are associated with the use of fossil fuels and the mining industry. An important source is currently represented by the remobilization and release of waste deposited in soils, sediments, water bodies, and garbage dumps; the incineration of municipal, medical and hazardous waste; and cremations and releases to the ground in cemeteries [40]. The maximum allowable limit for mercury (Hg) is 0.001 mg/L. The samples that exceed this limit are waterwheels 23N with a concentration of 0.004 mg/L and 25N with a concentration of 0.002 mg/L and wells 11P with a contraction of 0.002 mg/L, 18P with a concentration of 0.0015 mg/L, 20P with a concentration of 0.002 mg/L, and 22P with a concentration of 0.0012 mg/L (Figure 6c). From the distribution of the wells and waterwheels and the values recorded for these trace elements in the sampled waters reported here, it is inferred that they come from the leaching of sediments derived from the historical metallurgical processes of mining in Cerro de San Pedro due to the distribution of elements (Mn,As,Cd, and Hg) that were detected in this area. In the “Atlas de riesgo” for the municipalities of San Luis Potosí and Soledad de Graciano Sanchez (the northeastern part of the valley), a series of ground settlement fault segments are recorded, which may represent infiltration conduits for water into the deep aquifer. Given that the maximum concentration detected (0.004 mg/L) is from well 23N, and that well 20P located in the same area of the valley has a concentration of 0.002 mg/L (double that allowed by the standard), it can be argued that there is infiltration of this element from the shallow aquifer to the deep aquifer. However, for well 11P located in the NW of the VSLP, which is outside the influence of the zone of influence mentioned here, the possible contamination factor is the Peñazco municipal dump located in that sector given that land settlements also occur in that area, which is why it is known as the “Tierra Rajada” site (to NW of the valley). The most common damages occur in: the nervous system, affecting brain functions, which can cause degradation of the ability to learn, personality changes, tremors, vision changes, deafness, muscle incoordination, and memory loss; DNA and chromosomes; allergic reactions, skin irritation; tiredness; and headache; and negative effects on reproduction, with changes in sperm, birth defects, and miscarriage [40]. The detected values of the aforementioned trace elements represent contamination levels because they exceed the permissible limits of the NOM-127-SSA1-1994 [33] standard; these trace elements represent a danger to groundwater, nitrates, fecal coliforms, and total coliforms at concentrations lower than those reported [9,13]. In the superficial aquifer, there is evidence of infiltration of water from the unconfined aquifer into the deep free aquifer. These results corroborate that part of the nitrate, fecal coliform, and total and trace coliform that contaminated waters in the shallow unconfined aquifer as reported in previous studies [9,13] are infiltrating into the shallow unconfined type through the lattice of ground settlement faults in the valley. Extensive study of the waters of the deep aquifer wells allowed us to verify that the waters of the intermediate unconfined type are showing signs of anthropogenic, biogenic, and heavy metal contamination. This is of great importance, especially if the water from these wells is used for human consumption. The trace elements in the sampled waters reported here are inferred to come from the leaching of sediments derived from the historical metallurgical processes of mining in Cerro de San Pedro due to the distribution of elements (Mn,As,Cd, and Hg) detected in this area. It is assumed that there may be natural recharge of the deep aquifer in the northeastern margin of the valley, but the record of recent settlements that have been documented allows us to suppose that this process may also be recent or that its dilutional effect may be added if the first possibility has been active. That will require other studies not carried out here, which we can assume soon.

## 5. Conclusions

The current study reveals evidence of contamination resulting from anthropogenic activities in the deep unconfined type of the San Luis Potosí valley.

The deep aquifer has been infiltrated by organic contaminants, E. Coli, and total coliforms that are commonly found in the shallow aquifer. This contamination is especially prevalent in well 2P, which is situated in the area with the longest history of land settlement in the valley.Fats and oils, which are directly derived from anthropogenic activities and widely present in the shallow aquifer, have also been detected in the wells, suggesting that their infiltration into the deep aquifer is already underway.The nitrate levels in drinking water are due to contamination from animal waste or water spills from dairies or livestock, excessive use of fertilizers, and/or infiltration of human sewage from septic tanks.The presence of incipient and localized manganese (Mn) contamination in groundwater is caused by rainwater drainage in the limestone and dolomite that outcrop extensively in the Sierra de Alvarez, which borders the valley on its eastern side.Arsenic (As) in groundwater may be contributed by arsenic residues in the “Jales” derived from mining activity or occasional contributions of fertilizers containing arsenic.Three of the samples taken from the wells (11P, 12P, and 20P) showed cadmium levels two to four times higher than the allowable limit of the standard, which poses a high risk to human health.Mercury (Hg) concentrations in the wells suggests that contamination by this element comes from the leaching of sediments derived from historical metallurgical processes of mining in Cerro de San Pedro due to the distribution of elements Mn,As,Cd, and Hg (Figure 7).

The study suggests that there could be infiltration of contaminated water from the shallow aquifer into the deep aquifer through numerous subsurface faults in the central part of the valley. Initially considered unclear or insignificant, this issue is now proposed as a top priority due to its potential implications for both the population’s drinking water supply and public health in the short and long term. A more comprehensive investigation of the latter aquifer is deemed necessary to provide more thorough documentation of the reported findings.

## Figures and Tables

**Figure 1 ijerph-20-06152-f001:**
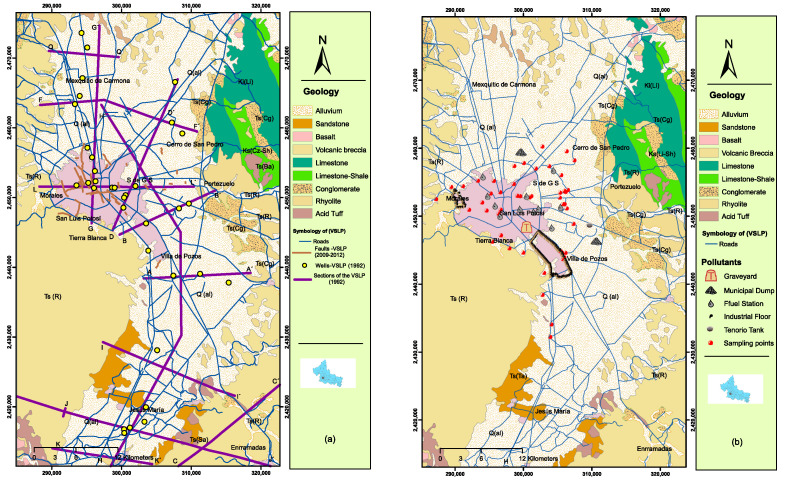
(**a**) Locations of subsurface geology sections based on lithologic columns from deep wells drilled in the valley (according to [11]) and faults (brown). These structures control land settlement in the valley and generally delineate the tectonic blocks covered by the alluvial fill (according to the “Atlas de riesgo of the municipalities of San Luis Potosí and Soledad de Graciano Sanchez 2018”). (**b**) Location of water samples from the unconfined aquifer reported by [9], wherein the water quality is documented based on its degree of contamination. (Arc map and surfer programs were used to create the images.)

**Figure 2 ijerph-20-06152-f002:**
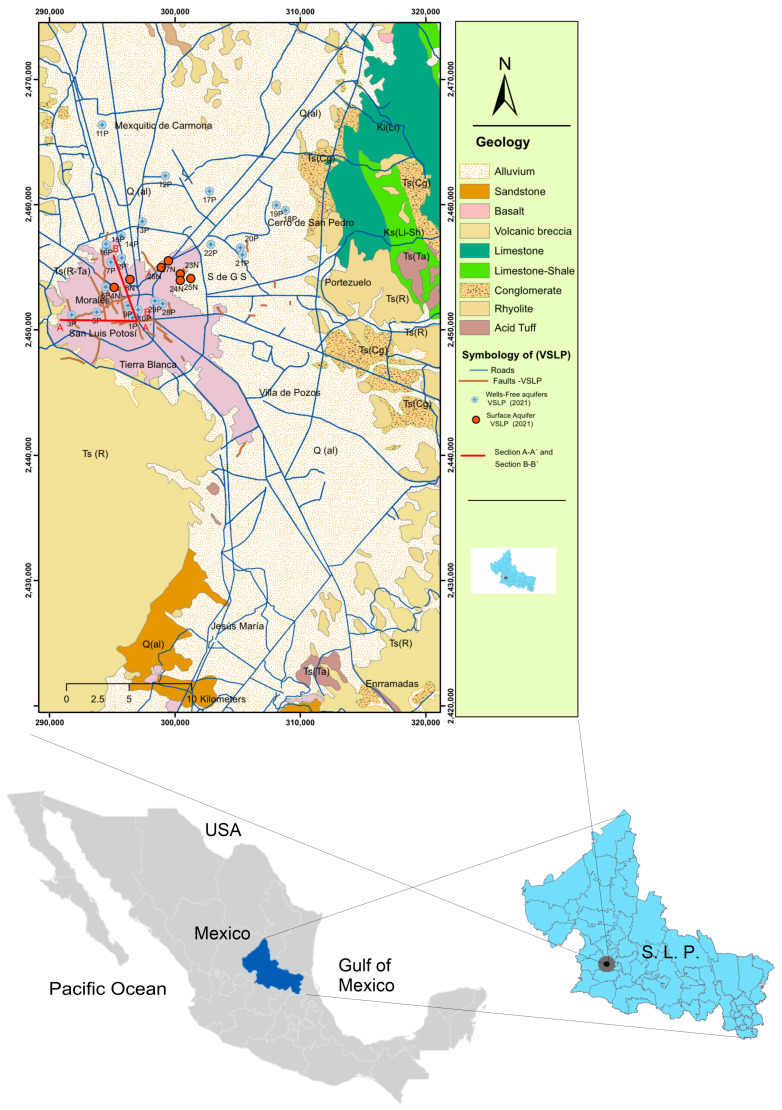
The San Luis Potosí aquifer is located in the central–western portion of the state of San Luis Potosí, has an area of about 1980 km${}^{2}$, and is between the municipalities of San Luis Potosí and Soledad de Graciano Sanchez in the state of San Luis Potosí. It is orographically limited to the north by a group of hills called “Alto La Melada”, to the west–south by the Sierra de San Miguelito, and to the east by the Sierra de Álvarez. (The arc map program was used to create the image).

**Figure 3 ijerph-20-06152-f003:**
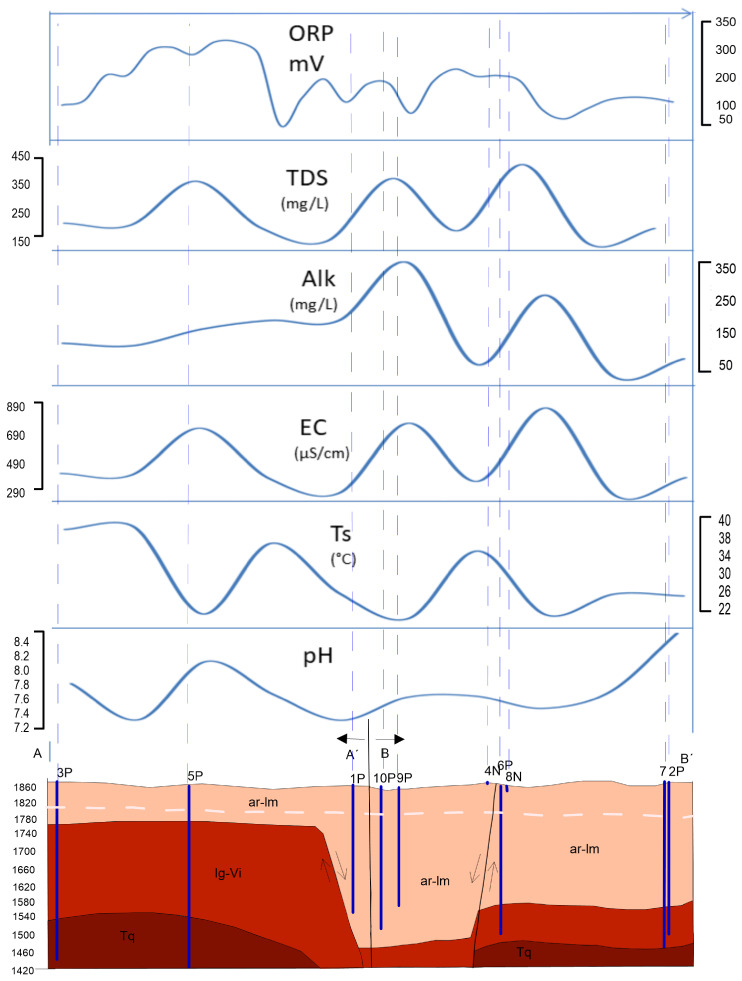
Schematic geological section of the San Luis Potosí Valley (VSLP). (Grapher 11 was used to create the image).

**Figure 4 ijerph-20-06152-f004:**
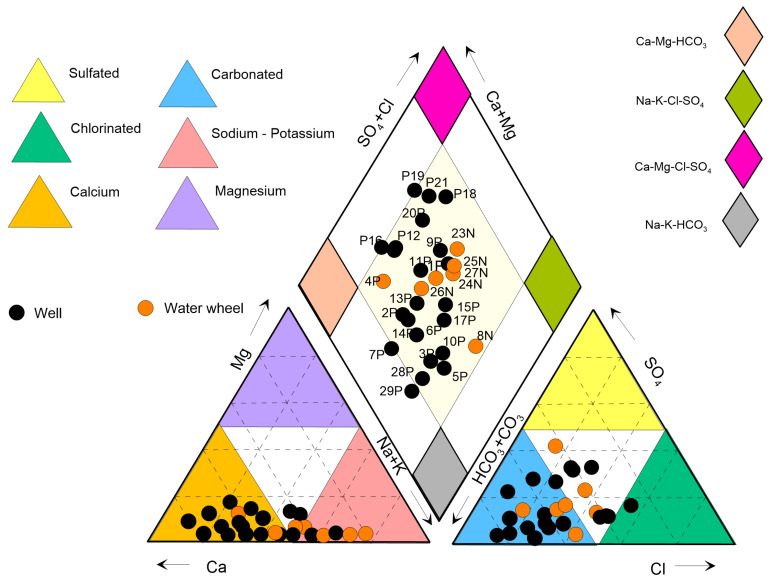
Piper diagram modified from [31] shows families of water. (Surfer was used to create the image).

**Figure 5 ijerph-20-06152-f005:**
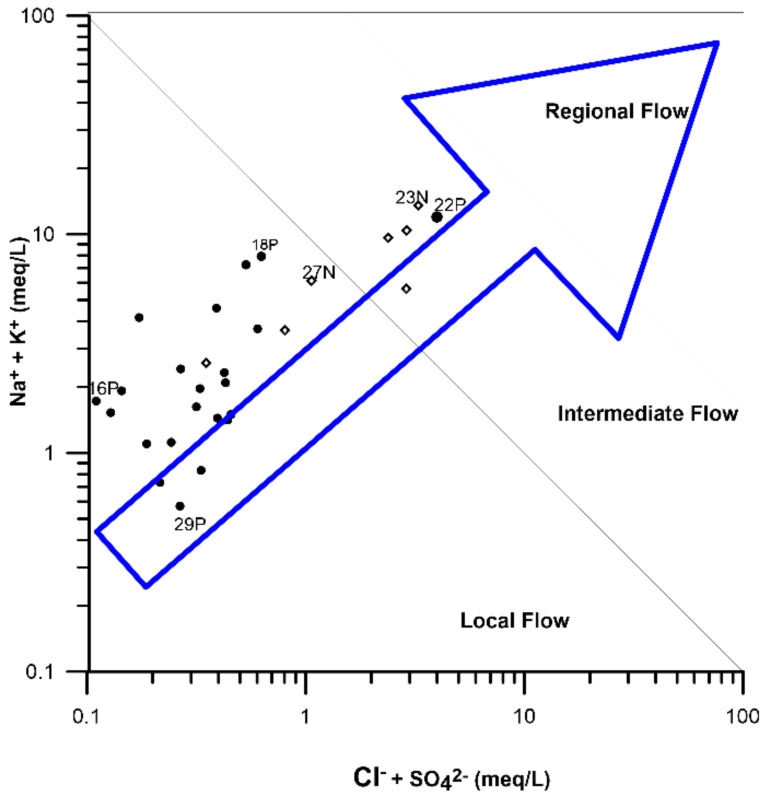
The Mifflin [32] diagram shows two different water flows, one local and one intermediate. (Grapher was used to create the image).

**Figure 6 ijerph-20-06152-f006:**
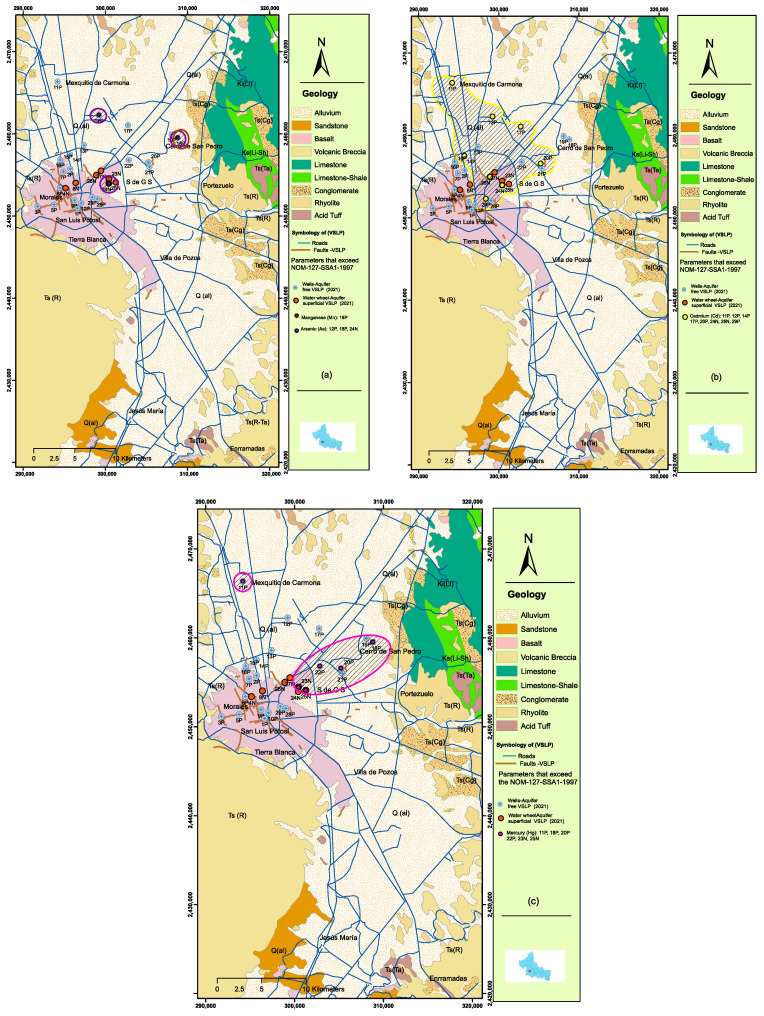
The shaded maps represent trace elements that exceed the allowable limits of the NOM-127-SSA1 [33] standard. Concentrations are indicated in (mg/L): (**a**) arsenic (As) concentration, (**b**) Cadmium (Cd) concentration, and (**c**) mercury (Hg) concentration. (Arc map software was used to create the image).

**Figure 7 ijerph-20-06152-f007:**
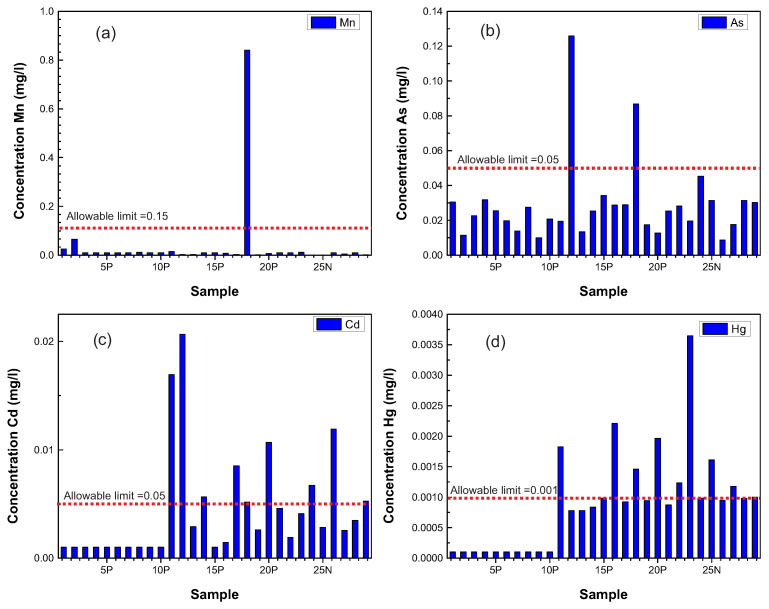
Concentrations of elements that exceed the allowable limits set by NOM-127-SSA1: (**a**) manganese (Mn) concentration, (**b**) arsenic (As) concentration, (**c**) cadmium (Cd) concentration, and (**d**) mercury (Hg) concentration. (Arc map program was used to create the image).

**Table 1 ijerph-20-06152-t001:** Physicochemical parameters of the in situ VSLP taken from the groundwater samples collected.

			UTM		Field Parameters							
					Ta	Ts	pH	EC	DO	TDS	Alk	ORP
Sample	Location	Sampling Point	Lat	Long	°C	°C		μS/cm	%	ppm		mV
1P	Alameda	well	296,631	2,450,966	23.0	22.8	8.0	769	24.0	391	189	118.6
2P	Saucito	well	295,751	2,455,752	23.0	26.0	8.5	377	31.7	183	119	132.0
3P	Parque Morales	well	291,769	2,451,178	31.0	37.7	7.7	410	-	206	156	209.0
4N	La Venadita	waterwheel	295,163	2,453,378	19.0	22.0	7.4	808	-	403	348	210.0
5P	Muñoz	well	293,763	2,451,413	33.0	38.2	7.0	395	15.4	197	150	284.0
6P	Tangamanga 2	well	294,471	2,453,418	30.0	33.9	7.4	347	32.5	173	105	295.5
7P	CNA-27	well	294,881	2,455,405	15.7	26.3	7.5	230	28.9	115	75	273.2
8N	3a.Chica	waterwheel	296,410	2,454,010	16.5	22.6	7.2	930	4.4	465	270	311.8
9P	República	well	296,244	2,451,936	20.0	26.4	7.0	253	32.5	127	210	312.0
10P	Montesillo	well	297,060	2,451,589	20.2	35.3	7.5	369	27.5	185	210	275.2
11P	Tierra Rajada	well	294,195	2,466,356	27.0	24.7	8.2	320	20.0	160	153	56.0
12P	Rinconada	well	299,229	2,462,282	27.0	26.0	7.8	335	60.0	162	165	140.0
13P	Milpillas	well	297,401	2,458,626	19.0	26.1	7.8	473	38.9	237	213	198.0
14P	Compostela	well	295,723	2,457,515	23.7	29.7	7.6	467	46.3	183	192	127.7
15P	Ponciano	well	294553	2456510	23.1	33.5	7.4	333	38.6	166	174	183.0
16P	Ponciano 1	well	294,510	2,456,843	25.0	31.5	7.5	306	41.5	153	153	183.5
17P	El Ranchito	well	302,757	2,461,064	22.7	23.5	7.0	298	57.2	145	129	93.9
18P	Well 2 Agronomia	well	308,813	2,459,533	25.0	23.5	6.7	1112	60.0	556	195	190.0
19P	Well 3 Agronomia	well	308,094	2,459,937	20.0	24.4	7.1	568	53.2	283	153	228.8
20P	Valle de la Palma	well	305,225	2,456,570	21.7	20.8	6.7	862	55.3	429	171	206.0
21P	El Diamante	well	305,381	2,456,010	18.4	20.4	6.6	1088	52.7	536	243	208.5
22P	El Zapote	well	302,858	2,456,810	18.0	20.5	6.9	1518	38.6	759	360	191.8
23N	Soledad 1	waterwheel	300,442	2,454,448	21.8	21.1	6.9	2156	8.2	1077	450	105.1
24N	Soledad 2	waterwheel	300,438	2,453,941	23.4	20.0	7.1	1914	12.2	944	390	76.5
25N	Soledad 3	waterwheel	301,278	2,454,092	24.5	20.5	7.1	1790	15.5	894	480	105.6
26N	Potrero de adentro	waterwheel	298,923	2,454,963	24.9	21.3	6.7	909	0.0	456	252	133.4
27N	Soledad 4	waterwheel	299,490	2,455,503	22.3	20.1	6.7	1028	5.1	514	393	142.4
28P	Well Praderas	well	299,024	2,452,087	22.9	33.2	7.3	279	30.0	138	162	139.7
29P	Well Las Palmas	well	298,407	2,452,303	25.4	34.8	7.5	297	30.3	149	138	128.0

Ambient temperature °C (TA), sample temperature °C (Ts), hydrogen potential (pH), electric conductivity (EC), dissolved oxygen (DO), total dissolved solids (TDS), alkalinity (Alk), and oxidation reduction potential (ORP).

**Table 2 ijerph-20-06152-t002:** Main characteristics of major elements and chemical families of groundwater.

Laboratory											
mg/L											
Sample	${\mathrm{Na}}^{+}$	$K^{+}$	${\mathrm{Ca}}^{+2}$	${\mathrm{Mg}}^{+2}$	${\mathrm{CO}}_{3}^{-2}$	${\mathrm{HCO}}_{3}^{-}$	${\mathrm{Cl}}^{-}$	${\mathrm{SO}}_{4}^{-2}$	$N-{\mathrm{NO}}_{3}^{-}$	$F^{-}$	%Error
1P	49.8	44.6	59.4	9.9	0.0	212.4	60.9	92.0	7.8	0.0	−3.9
2P	29.6	8.6	36.4	1.7	0.0	146.0	30.5	11.0	4.2	0.1	−4.4
3P	61.0	3.4	24.6	3.7	0.0	186.7	10.2	55.0	2.5	0.5	−4.7
4N	50.0	5.4	99.8	9.3	0.0	340.4	45.7	60.0	7.6	0.6	−3.7
5P	60.0	2.6	20.0	1.9	0.0	153.7	30.5	25.0	2.4	2.1	−4.4
6P	39.0	4.5	27.6	3.7	0.0	142.7	25.4	19.0	2.5	1.7	−3.0
7P	27.0	19.2	33.8	2.8	0.0	164.7	20.3	7.0	3.0	0.7	−1.1
8N	196.0	35.6	52.2	4.7	0.0	318.4	116.8	105.0	15.0	1.0	2.5
9P	24.0	5.4	24.6	2.8	0.0	65.9	40.6	16.0	3.1	1.9	−4.3
10P	53.0	8.2	23.0	1.9	0.0	142.7	30.5	27.0	2.7	1.3	−2.6
11P	37.5	11.2	53.4	1.9	0.0	151.3	67.5	14.4	2.1	0.6	−1.3
12P	24.8	7.5	68.3	2.0	0.0	156.2	49.6	22.5	0.5	0.6	3.8
13P	41.2	16.6	48.7	2.1	0.0	190.3	49.6	25.0	0.7	0.7	−3.0
14P	44.3	9.3	47.4	1.4	0.0	200.1	41.7	20.0	0.3	1.2	−3.2
15P	57.4	5.4	35.0	1.0	0.0	146.4	53.6	35.0	1.7	2.2	−4.4
16P	16.6	13.4	65.3	5.2	0.0	161.0	39.7	27.5	1.9	2.2	1.8
17P	57.5	5.3	36.7	0.5	0.0	151.3	33.8	55.0	1.8	0.4	−2.3
18P	65.7	22.2	115.8	15.0	0.0	185.4	210.4	87.5	5.0	0.5	−3.4
19P	25.6	12.7	86.1	16.0	0.0	151.3	111.2	45.0	0.0	0.6	3.5
20P	41.7	26.5	93.5	15.1	0.0	195.2	95.3	90.0	5.0	0.4	1.7
21P	53.2	29.7	137.9	19.2	0.0	224.5	123.1	175.0	5.0	0.4	1.6
22P	267.9	0.2	31.1	0.1	0.0	258.6	200.5	200.0	5.0	0.6	−4.4
23N	208.6	41.4	182.7	26.1	0.0	463.6	234.3	300.0	5.0	0.6	1.4
24N	194.3	37.7	161.2	24.3	0.0	453.8	95.3	365.0	5.0	0.0	3.7
25N	172.7	31.1	146.9	22.5	0.0	400.2	152.9	250.0	5.0	0.0	3.2
26N	72.1	34.4	76.1	16.4	0.0	278.2	43.7	112.5	5.0	2.0	3.2
27N	89.9	36.8	81.1	18.3	0.0	258.6	85.4	175.0	5.0	2.0	−1.6
28P	47.5	6.2	25.6	1.6	0.0	180.6	23.8	7.0	0.6	1.2	−3.3
29P	42.2	3.6	22.2	0.6	0.0	151.3	13.9	8.0	0.5	1.4	−1.0

**Table 3 ijerph-20-06152-t003:** The results of nitrate nitrogen ($N-NO_{3}^{-}$) and phosphorus (*P*).

	Laboratory	
	$N-{\mathrm{NO}}_{3}^{-}$	P
**Sample**	**mg/L**	**mg/L**
1P	7.8	0.216
2P	4.2	<0.01
3P	2.5	0.002
4N	7.6	0.131
5P	2.4	0.003
6P	2.5	0.019
7P	3	0.085
8N	15	0.449
9P	3.1	0.261
10P	2.7	0.007
11P	2.1	<0.01
12P	0.5	<0.01
13P	0.70	<0.01
14P	0.30	<0.01
15P	1.7	<0.01
16P	1.9	<0.01
17P	1.8	<0.01
18P	5	0.015
19P	0	<0.01
20P	5	<0.01
21P	5	<0.01
22P	5	0.087
23N	5	<0.01
24N	5	0.158
25N	5	0.011
26N	5	<0.01
27N	5	0.069
28P	0.6	<0.01
29P	0.5	<0.01

Maximum permissible limits: nitrogen from nitrates $N-NO_{3}^{-}=10$ mg/L, NOM-127-SSA1-1994; phosphorus of phosphates $PO_{4}=0.1$ mg/L, WHO-2004; and total phosphorus PT = 5 mg/L, NOM-001-SSA1-1996.

**Table 4 ijerph-20-06152-t004:** Bacteriological concentrations: permissible limit for Fecal Col. = 0 most probable number per 100 (0 NMP/100) not detectable, NOM-127; permissible limit for Total Col. = (2NMP/100), NOM-127; and permissible limit for Fats and Oils (G and A) = (21 mg/L), NOM-002.

	Fecal	Total	Fats and	Depth
Sample	Coliforms	Coliforms	Oils (mg/L)	(m)
1P	<3	<3	6.63	300
2P	<3	150	2.65	360
3P	8	8	5.1	420
4N	<3	<3	3.7	3
5P	<3	<3	2.9	200
6P	<3	<3	6.8	418
7P	<3	<3	4.4	400
8N	<3	<3	6.8	5
9P	<3	<3	7.7	280
10P	<3	<3	7.7	700
11P	<3	<3	-	200
12P	11	13	-	350
13P	<3	7	-	250
14P	<3	<3	-	360
15P	11	11	-	360
16P	11	16	-	400
17P	14	14	-	350
18P	5	10	-	350
19P	17	17	-	222
20P	11	16	-	70
21P	<3	<3	-	60
22P	5	7	-	1180
23N	<3	<3	-	8
24N	11	11	-	10
25N	<3	<3	-	7
26N	<3	<3	-	5
27N	5	5	-	10
28P	11	11	-	600
29P	<3	<3	-	300

Permissible limits of fecal coliforms = no detectables (0 NMP/100 mL), NOM-127; permissible limits of total coliforms = (2 NMP/100 mL), NOM-127; and permissible limits of fats and oil = (21 mg/L), NOM-001.

**Table 5 ijerph-20-06152-t005:** Concentrations of heavy metals and trace element analysis. Permissible limit in (mg/L or ppm) of: aluminum (Al) = 0.2, chromium (Cr) = 0.05, manganese (Mn) = 0.15, iron (Fe) = 0.3, copper (Cu) = 2, zinc (Zn) = 5, and arsenic (As), cadmium (Cd), and mercury (Hg) = 0.001 according to NOM-127-SSA1-1994.

Sample	Li	B	Al	P	Sc	Ti	V	Cr	Mn	Fe	Ni	Cu	Zn	Ga	Ge
	ppm	ppm	ppm	ppm	ppm	ppm	ppm	ppm	ppm	ppm	ppm	ppm	ppm	ppm	ppm
1P	0.00	0.14	0.004	0.216	0.006	0.004	0.008	0.0002	0.026	<0.01	0.0007	<0.01	0.01	0.002	<0.01
2P	0.07	0.09	0.021	<0.01	0.009	0.005	0.005	<0.01	0.065	<0.01	<0.01	<0.01	<0.01	0.002	<0.01
3P	0.26	0.24	<1.0	0.002	0.005	0.002	0.000	<0.01	<0.01	0.009	<0.01	<0.01	<0.01	<0.01	0.002
4N	0.01	0.22	0.062	0.13	0.010	0.006	0.016	0.0005	<0.01	0.005	<0.01	<0.01	0.00	0.002	<0.01
5P	0.27	0.26	<1.0	0.00	0.005	0.002	0.000	<0.01	<0.01	0.000	<0.01	<0.01	<0.01	<0.01	0.002
6P	0.17	0.17	0.031	0.02	0.007	0.004	0.002	<0.01	<0.01	0.016	<0.01	<0.01	<0.01	<0.01	0.001
7P	0.03	0.07	<1.0	0.09	0.009	0.006	0.005	<0.01	<0.01	0.001	<0.01	<0.01	<0.01	0.000	<0.01
8N	0.01	0.24	0.009	0.45	0.010	0.006	0.012	0.003	0.011	0.006	<0.01	0.0003	0.00	0.001	<0.01
9P	0.02	0.07	<1.0	0.26	0.010	0.006	0.005	<0.01	<0.01	<0.01	<0.01	<0.01	<0.01	0.001	<0.01
10P	0.00	0.21	<1.0	0.007	0.005	0.002	0.001	<0.01	<0.01	<0.01	<0.01	0.0001	0.00	<0.01	0.001
11P	0.07	0.14	0.075	<0.01	0.006	0.004	0.008	0.002	0.015	0.114	0.008	0.017	0.21	<0.01	<0.01
12P	0.26	0.09	0.024	<0.01	0.009	0.005	0.005	0.008	0.003	0.099	0.009	0.001	0.13	<0.01	<0.01
13P	0.01	0.24	0.027	<0.01	0.005	0.002	0.000	0.004	0.002	0.172	0.001	<0.01	0.17	<0.01	<0.01
14P	0.27	0.22	0.028	<0.01	0.010	0.006	0.016	0.001	<0.01	0.052	0.000	0.012	0.21	<0.01	<0.01
15P	0.17	0.26	0.002	<0.01	0.005	0.002	0.000	<0.01	< 0.01	0.025	<0.01	<0.01	0.13	<0.01	<0.01
16P	0.03	0.17	0.027	<0.01	0.007	0.004	0.002	<0.01	0.007	0.261	0.002	0.010	0.17	<0.01	<0.01
17P	0.01	0.07	0.016	<0.01	0.009	0.006	0.005	0.005	0.002	0.061	0.006	<0.01	0.21	<0.01	<0.01
18P	0.02	0.24	0.057	0.015	0.010	0.006	0.012	0.003	0.841	0.233	0.016	0.005	0.13	<0.01	<0.01
19P	0.00	0.07	0.044	<0.01	0.010	0.006	0.005	0.000	0.001	0.048	0.003	<0.01	0.17	<0.01	<0.01
20P	0.07	0.21	0.031	<0.01	0.005	0.002	0.001	0.000	0.006	0.059	0.003	0.017	0.21	<0.01	<0.01
21P	0.26	0.14	0.005	<0.01	0.006	0.004	0.008	0.003	<0.01	0.063	0.008	<0.01	0.13	<0.01	<0.01
22P	0.01	0.09	0.009	0.087	0.009	0.005	0.005	0.001	<0.01	0.097	0.009	<0.01	0.17	<0.01	<0.01
23N	0.27	0.24	0.067	<0.01	0.005	0.002	0.000	0.001	0.012	0.156	0.008	<0.01	0.21	<0.01	<0.01
24N	0.17	0.22	0.008	0.158	0.010	0.006	0.016	0.004	0.000	0.066	0.011	<0.01	0.13	<0.01	<0.01
25N	0.03	0.26	0.000	0.011	0.005	0.002	0.000	<0.01	0.001	0.042	0.006	<0.01	0.17	<0.01	<0.01
26N	0.01	0.17	0.028	<0.01	0.007	0.004	0.002	<0.01	<0.01	0.032	0.002	<0.01	0.21	<0.01	<0.01
27N	0.02	0.07	0.054	0.069	0.009	0.006	0.005	0.0001	0.006	0.067	0.003	<0.01	0.13	<0.01	<0.01
28P	0.00	0.24	0.022	<0.01	0.010	0.006	0.012	0.0018	<0.01	0.030	0.002	<0.01	0.17	<0.01	<0.01
29P	0.07	0.07	0.033	<0.01	0.010	0.006	0.005	0.0026	0.001	0.055	0.003	<0.01	0.21	<0.01	<0.01
Limit *				0.2					0.05	0.15	0.3		2	5	

* Maximum allowable limit according to NOM-127-SSA1-1994 [33].

**Table 6 ijerph-20-06152-t006:** Concentrations of heavy metals and trace element analysis. Permissible limit in (mg/L or ppm) of: aluminum (Al) = 0.2, chromium (Cr) = 0.05, manganese (Mn) = 0.15, iron (Fe) = 0.3, copper (Cu) = 2, zinc (Zn) = 5, and arsenic (As), cadmium (Cd), and mercury (Hg) = 0.001 according to NOM-127-SSA1-1994.

Sample	As	Se	Br	Rb	Sr	Mo	Ag	Cd	I	Cs	Ba	U	W	Hg
	ppm	ppm	ppm	ppm	ppm	ppm	ppm	ppm	ppm	ppm	ppm	ppm	ppm	ppm
1P	0.03	0.001	0.207	0.116	0.580	0.005	<0.01	<0.01	0.063	0.001	0.150	0.009	<0.01	<0.01
2P	0.01	0.000	0.124	0.029	0.244	0.000	<0.01	<0.01	0.052	0.011	0.139	0.010	<0.01	<0.01
3P	0.02	0.002	0.213	0.036	0.067	0.001	<0.01	<0.01	0.050	0.025	0.006	0.008	<0.01	<0.01
4N	0.03	0.004	0.206	0.128	0.601	0.001	<0.01	<0.01	0.065	<0.01	0.147	0.012	<0.01	<0.01
5P	0.03	0.002	0.209	0.031	0.071	0.001	<0.01	<0.01	0.050	0.027	0.010	0.007	<0.01	<0.01
6P	0.02	0.002	0.165	0.033	0.090	0.000	<0.01	<0.01	0.046	0.018	0.019	0.005	<0.01	<0.01
7P	0.01	0.001	0.112	0.029	0.122	0.000	<0.01	<0.01	0.025	0.012	0.055	0.000	<0.01	<0.01
8N	0.03	0.007	0.423	0.104	0.404	0.002	<0.01	<0.01	0.070	<0.01	0.089	0.019	<0.01	<0.01
9P	0.01	0.002	0.134	0.051	0.168	0.000	<0.01	<0.01	0.033	0.014	0.106	0.001	<0.01	<0.01
10P	0.02	0.002	0.195	0.027	0.069	0.001	<0.01	<0.01	0.045	0.018	0.005	0.007	<0.01	<0.01
11P	0.02	0.001	0.207	0.116	0.580	0.617	<0.01	0.017	0.063	1.252	84.0	0.010	0.000	0.002
12P	0.13	0.000	0.124	0.029	0.244	<0.01	<0.01	0.021	0.052	5.407	24.7	0.008	0.000	0.001
13P	0.01	0.002	0.213	0.036	0.067	<0.01	<0.01	0.003	0.050	<0.01	95.0	0.012	0.000	0.001
14P	0.03	0.004	0.206	0.128	0.601	0.000	<0.01	0.006	0.065	9.081	57.8	0.007	0.000	0.001
15P	0.03	0.002	0.209	0.031	0.071	0.001	<0.01	<0.01	0.050	10.835	20.2	0.005	0.000	0.001
16P	0.03	0.002	0.165	0.033	0.090	<0.01	<0.01	0.001	0.046	<0.01	69.4	0.000	0.000	0.002
17P	0.03	0.001	0.112	0.029	0.122	0.000	<0.01	0.009	0.025	9.706	11.6	0.019	0.000	0.001
18P	0.09	0.007	0.423	0.104	0.404	<0.01	<0.01	0.005	0.070	<0.01	101.6	0.001	0.000	0.0015
19P	0.02	0.002	0.134	0.051	0.168	<0.01	<0.01	0.003	0.033	<0.01	52.7	0.007	0.000	0.001
20P	0.01	0.002	0.195	0.027	0.069	<0.01	<0.01	0.011	0.045	<0.01	75.2	0.010	0.000	0.002
21P	0.03	0.001	0.207	0.116	0.580	<0.01	<0.01	0.005	0.063	<0.01	49.7	0.008	0.000	0.001
22P	0.03	0.000	0.124	0.029	0.244	<0.01	<0.01	0.002	0.052	<0.01	81.9	0.012	0.000	0.0012
23N	0.02	0.002	0.213	0.036	0.067	<0.01	<0.01	0.004	0.050	<0.01	30.2	0.007	0.000	0.004
24N	0.05	0.004	0.206	0.128	0.601	<0.01	<0.01	0.007	0.065	<0.01	28.0	0.005	0.000	0.001
25N	0.03	0.002	0.209	0.031	0.071	<0.01	<0.01	0.003	0.050	<0.01	27.9	0.000	0.000	0.002
26N	0.01	0.002	0.165	0.033	0.090	0.002	<0.01	0.012	0.046	<0.01	44.6	0.019	0.000	0.001
27N	0.02	0.001	0.112	0.029	0.122	0.004	<0.01	0.003	0.025	<0.01	56.5	0.001	0.000	0.001
28P	0.03	0.007	0.423	0.104	0.404	0.002	<0.01	0.003	0.070	9.252	5.8	0.007	0.000	0.001
29P	0.03	0.002	0.134	0.051	0.168	0.004	<0.01	0.005	0.033	7.538	13.1	0.010	0.000	0.001
Limit *	0.05							0.005			0.7			0.001

* Maximum allowable limit according to NOM-127-SSA1-1994 [33].

## Data Availability

The data generated in this study are presented in the tables and maps in this manuscript.

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
