# Peer review of "Anthropogenic Contamination in the Free Aquifer of the San Luis Potosí Valley"

_ijerph, 2023, doi:10.3390/ijerph20126152_

Round 1

Reviewer 1 Report

The text highlights the serious issue of groundwater contamination in the San Luis Potosí valley and its potential impact on public health and the availability of safe drinking water. The contamination of the shallow aquifer is of particular concern as it is now affecting the deep free aquifer, which serves as a vital source of drinking water for the population. The presence of biogenic and potentially toxic trace elements, along with high levels of bacteriological indicators and nitrates, underscores the urgency of addressing this problem. The text emphasizes the need for immediate attention to prevent further contamination and mitigate the potential health risks associated with the consumption of contaminated water.

The topic of the article is interesting and stimulating. Here you can find my comments:

OVERALL: - Please, enlarge and re-arrange font sizes to guide the reader properly in all sections. All figures must be composed of HD images. It is mandatory to improve the scientific quality of the whole manuscript. Please, pay attention to the JOURNAL TEMPLATE within the entire manuscript: in all sections, including tables, references, captions, units, equations, and Figures. Please improve the references of the introduction to increase the scienfitic sound of it.

Line 1: "hanging aquifer" should be "unconfined aquifer."

Line 5: "coliformes fecales" should be "fecal coliforms."

Line 7: "mayor's" should be "major."

Line 16: Add references

Line 18: add references

Line 18-22: add references

Line 38: "assigned to" should be "attributed to”.

Line 47: should be (Figure 1b)

Line 62: "free type" should be "unconfined type."

Line 462-484: Conclusions should be summarized with bullet points. 

The quality of English in the paper is acceptable, with clear communication of ideas and appropriate use of language. However, there are some minor revisions needed to enhance the scientific soundness of the paper.

Firstly, it is important to ensure consistency in terminology and notation throughout the paper. Review the text to identify any inconsistencies or discrepancies in the use of terms, symbols, and abbreviations. This will improve clarity and prevent any confusion for the readers.

Secondly, pay attention to the organization and structure of the paper. Ensure that the information is presented in a logical sequence, with clear headings and subheadings to guide the reader. Check the flow of ideas within paragraphs and between sections, ensuring that each paragraph contributes to the overall coherence of the paper.

Thirdly, review the accuracy and precision of the scientific information presented. Verify the data, calculations, and references to ensure their correctness and relevance to the study. Any assumptions or limitations should be clearly stated and discussed, providing a comprehensive understanding of the research.

Additionally, consider the appropriate use of scientific language and terminology. Use precise and concise language to convey the research findings, avoiding unnecessary jargon or excessive technical details. Ensure that definitions of specialized terms are provided when needed, especially for readers who may be less familiar with the subject matter.

Lastly, proofread the paper for grammar, spelling, and punctuation errors. These minor revisions will help improve the overall readability and professionalism of the paper.

In summary, while the quality of English in the paper is acceptable, incorporating these minor revisions will enhance the scientific soundness of the paper. By ensuring consistency, improving organization and structure, verifying accuracy, using appropriate scientific language, and proofreading for errors, the paper will become even more effective in communicating its research findings.

Reviewer 2 Report

Minor editing of English language required.

Reviewer 3 Report

This manuscript aimed to investigate possible anthropogenic contamination of aquifers from The San Luis Potosí Valley. The work is of broad interest, and presents important information about anthropogenic contamination of these ground waters alerting for possible risk for human consumers. The work is well performed and designed. Data is presented and well discussed. There are only some minor corrections sent to the authors.

Line 149: please correct 0.45 μ to 0.45 μm

Line 154: “concentrated nitric acid ???3” please give values and references

Line 180-181. Authors state that T starts at 38ºC (please add information in the text to what sampling point corresponds) and increases up to 26ºC. but this is a decrease, please correct or clarify as T oscillate.

Line 270-271: please refer de metals and trace elements by symbol

Round 2

Reviewer 2 Report

The manuscript is now much improved. The authors have done a great job of responding to comments all over the article and improving the text as well. This manuscript will make a good contribution to the literature as a research article. Thank you for all your efforts. I recommend to accept this article to publish. 

Minor editing of English language required